Prey selectivity and the effect of diet on growth and development of a dragonfly, Sympetrum sanguineum

Dudová Pavla 1 2
http://orcid.org/0000-0001-8181-7458 Boukal David S. 1 3
http://orcid.org/0000-0003-0633-3896 Klecka Jan 1 jan.klecka@entu.cas.cz
1 Czech Academy of Sciences, Biology Centre, Institute of Entomology , České Budějovice , Czech Republic
2 Department of Zoology, Faculty of Science, University of South Bohemia , České Budějovice , Czech Republic
3 Department of Ecosystem Biology & Soil and Water Research Infrastructure, Faculty of Science, University of South Bohemia , České Budějovice , Czech Republic
Gillespie Joseph
Electronic publication date: 2019 Nov 5
Publication date: 2019
Volume: 7
Electronic Location ID: e7881
Received 2019 May 7; Accepted 2019 Sep 12
Copyright: © 2019 Dudová et al.
Copyright year: 2019
Copyright holder: Dudová et al.
License: This is an open access article distributed under the terms of the Creative Commons Attribution License, which permits unrestricted use, distribution, reproduction and adaptation in any medium and for any purpose provided that it is properly attributed. For attribution, the original author(s), title, publication source (PeerJ) and either DOI or URL of the article must be cited.
License URL: https://creativecommons.org/licenses/by/4.0/

Keywords: Predation, Predator-prey interactions, Prey selection, Switching, Optimal foraging theory, Ecological stoichiometry, Nutritional ecology, Survival, Aquatic insects

Funding: University of South Bohemia, Faculty of Science, Department of Ecosystem Biology & Soil and Water Research Infrastructure (MEYS; projects LM2015075 and EF16_013/0001782—SoWa Ecosystems Research) The work of David S. Boukal was supported by University of South Bohemia, Faculty of Science, Department of Ecosystem Biology & Soil and Water Research Infrastructure (MEYS; projects LM2015075 and EF16_013/0001782—SoWa Ecosystems Research). The funders had no role in study design, data collection and analysis, decision to publish, or preparation of the manuscript.

==============================
Despite a long tradition of research, our understanding of mechanisms driving prey selectivity in predatory insects is limited. According to optimal foraging theory, predators should prefer prey which provides the highest amount of energy per unit time. However, prey selectivity may also depend on previous diet and specific nutritional demands of the predator. From the long-term perspective, diet composition affects predator fitness. An open question is whether short-term selectivity of predators provides a diet which is optimal in the long-term. To shed more light on these issues, we conducted laboratory experiments on prey selectivity and its long-term consequences in larvae of the dragonfly Sympetrum sanguineum. We conditioned the larvae to one of two prey types, the cladoceran Daphnia magna and larvae of a non-biting midge Chironomus sp., and then exposed them to various combinations of the two prey types. We found that dragonfly larvae conditioned to Chironomus larvae consumed the same amount of D. magna, but significantly less Chironomus larvae compared to dragonfly larvae conditioned to D. magna. However, there was no effect of previous diet on their success of capture and handling time, suggesting a limited role of learning in their ability to process prey. We then tested the long-term effects of diets with different proportions of both prey for survival and growth of the dragonfly larvae. Individuals fed Chironomus-only diet had higher mortality and slower growth than dragonflies fed D. magna, while larvae fed a mixed diet had the highest survival and growth rate. In conclusion, we show that dragonfly larvae fed by Chironomus larvae performed poorly and compensated by preferring D. magna when both prey types were available. The superiority of the mixed diet suggests that a diverse diet may be needed to satisfy nutritional demands in S. sanguineum larvae. We demonstrate that merging short-term predation experiments with relevant data on predator fitness may provide better understanding of predator-prey interactions and conclude that detailed information on the (mis)matches between prey composition and predator nutritional demands is needed for further progress.

Introduction

Predators do not attack any prey indiscriminately, but feed more or less selectively on a subset of prey they encounter (Waldbauer & Friedman, 1991; Klecka & Boukal, 2012). Early research into the mechanisms and consequences of selective predation centred around the concept of optimal foraging theory (Emlen, 1966; MacArthur & Pianka, 1966; Stephens & Krebs, 1986) positing that consumers should maximise their energy intake by selectively consuming the most profitable resource, that is, a resource which provides the highest energy intake per unit of time. Evidence supporting optimal foraging theory started to accumulate from different consumer types including predators, herbivores, and parasitoids (Stephens & Krebs, 1986). At the same time, the theory was criticised for being simplistic among other reasons (Pyke, 1984; Pierce & Ollason, 1987). Reviews of experimental evidence have been inconclusive, because the level of support for predictions of optimal foraging theory varies widely between studies (Pyke, 1984; Stephens & Krebs, 1986; Sih & Christensen, 2001). Despite that, the appeal of optimal foraging theory as a mechanistic underpinning of selective predation has been bolstered by studies demonstrating that its predictions could be used to fit the structure of empirically observed food webs (Beckerman, Petchey & Warren, 2006; Petchey et al., 2008). However, more detailed understanding of nutritional demands of a growing number of species paints a more complex picture of mechanisms driving selective foraging (Fagan et al., 2002; Raubenheimer, Simpson & Mayntz, 2009; Wilder & Eubanks, 2010; Lefcheck et al., 2013) with implications for population and community dynamics (Moe et al., 2005) and nutrient cycling in freshwater ecosystems (Brett et al., 2017).

Predators do not use prey only as a source of energy, but also to obtain building blocks for their tissues (Raubenheimer, Simpson & Mayntz, 2009). The nutrient composition of prey (proteins, lipids, carbohydrates), its elemental composition such as the C:N:P ratio, or essential micronutrient content such as the long-chain polyunsaturated fatty acids (Guo et al., 2018) relative to the demands of a predator thus provides a more realistic basis for studies of selective predation. While Ecological Stoichiometry (Elser et al., 2000; Sterner & Elser, 2002) has shown that elemental composition of predators and prey often differs, which suggests that the growth and fitness of predators may be limited by individual elements (Fagan et al., 2002), growing research on nutritional ecology, exemplified by the Geometric Framework (Raubenheimer, Simpson & Mayntz, 2009), shows that specific macronutrients, rather than elements, determine the performance of predators fed different diets (Raubenheimer, Simpson & Mayntz, 2009; Wilder & Eubanks, 2010). Specifically, while lipids serve as a source of energy, proteins are needed to build up tissues. Also, digestibility of prey depends on its composition in terms of macromolecules rather than energetic contents or elemental composition. For example, the exoskeleton in invertebrates contains a large amount of nitrogen which is indigestible, unlike nitrogen in proteins (Wilder, Barnes & Hawlena, 2019). Relative lipid and protein content of prey affects physiology, behaviour, and fitness of predators (Jensen et al., 2012; Schmidt et al., 2012; Koemel, Barnes & Wilder, 2019), and predators can feed selectively to reach their target intake of lipids and proteins (Jensen et al., 2012). Recent work on long-chain poly-unsaturated fatty acids further identified micronutrient compounds that are essential for invertebrate development and reproduction (Brett et al., 2017; Guo et al., 2018).

Insights from both Ecological Stoichiometry (Elser et al., 2000; Sterner & Elser, 2002) and the Geometric Framework for nutrition (Raubenheimer, Simpson & Mayntz, 2009; Wilder & Eubanks, 2010) suggest that predators could rarely depend on a single resource to reach their desired intake target of different elements or (micro)nutrients and may be able to adjust their feeding on different prey types accordingly (Pulliam, 1975; Raubenheimer & Simpson, 1997; Mayntz et al., 2005; Jensen et al., 2012; Brett et al., 2017). Indeed, previous experiments on diet-dependent growth and reproduction of animals have found that species often perform best on mixed diets (Lefcheck et al., 2013). Mixed diet may not only be nutritionally balanced compared to single prey type, but it may also help deal with toxic prey by dilution of toxins (Freeland & Janzen, 1974; Bernays et al., 1994). Rejecting toxic or unpalatable prey seems trivial, but in some cases predators do not learn to avoid toxic prey even when they have alternative prey to rely upon (Oelbermann & Scheu, 2002). However, the empirical support of the toxin-dilution hypothesis is weak and the balanced-diet hypothesis is better supported by experimental data (Lefcheck et al., 2013). Finally, most production in freshwater food webs depends on microalgae due to their highly favourable biochemical composition (reviewed in Brett et al. (2017)), and predator diets should thus include consumers that feed on such microalgae.

Studies of predator-prey interactions usually take either a behavioural approach based on short-term experiments, or focus on growth, reproduction, and population dynamics at a longer time scale. While short-term experiments of foraging behaviour help to elucidate the process of prey search, capture, and processing (Lawton, Beddington & Bonser, 1974; Sih & Christensen, 2001; Klecka & Boukal, 2014), long-term experiments are needed to examine implications of diet for fitness of consumers and evolution of interspecific interactions (Moe et al., 2005; Lefcheck et al., 2013). However, these two approaches have rarely been combined in a single study system.

We used larvae of the dragonfly Sympetrum sanguineum Müller, 1764 (Odonata: Libellulidae) to investigate the links between short-term foraging decisions and long-term fitness consequences. Despite their popularity in freshwater ecology studies, mechanistic basis of prey selectivity in dragonfly larvae is little understood. As other predators, they are at least partly size-selective (Hirvonen & Ranta, 1996; Turner & Chislock, 2007; Klecka & Boukal, 2013), influenced by the behaviour and microhabitat use of their own and potential prey (Cooper, Smith & Bence, 1985; Johansson, 1993; Klecka & Boukal, 2012, 2013), and their predation is modulated by habitat structure (Klecka & Boukal, 2014). Evidence of learning capacity in dragonfly larvae in a foraging context suggests that they may learn to avoid unpalatable prey (Rowe, 1994) and handle prey more efficiently based on previous experience (Bergelson, 1985). Frequency-dependent food selection and prey switching (Lawton, Beddington & Bonser, 1974; Sherratt & Harvey, 1993), that is, disproportionate preference of abundant prey and avoidance of rare prey, was also reported in larvae of a damselfly (Akre & Johnson, 1979) and a dragonfly (Bergelson, 1985). While dragonfly and damselfly larvae are commonly used in short-term predation experiments, data on long-term consequences of diet composition for their growth, survival, and reproduction are very limited (Baker, 1989; Hottenbacher & Koch, 2006).

We conducted two experiments to address short-term prey selectivity and long-term effects of diet on the growth and survival of S. sanguineum larvae. We tested whether and how their prey selectivity depends on relative abundances of two prey types, Daphnia magna Straus, 1820 (Cladocera: Daphniidae) and larvae of Chironomus sp. (Diptera: Chironomidae) and on their previous diet. We hypothesised that conditioning the predator to one of the prey types would increase its preference for that prey in the experiment because the predator would learn to efficiently capture and handle that prey. Alternatively, the predator could preferentially select the other prey type if the single-prey diet during the conditioning period was not nutritionally balanced. To corroborate the findings from the short-term experiment, we conducted a long-term experiment to test how diet composition affects survival and growth of S. sanguineum larvae. Here, we hypothesised that S. sanguineum larvae would survive and grow best on a mixed diet which could more completely satisfy their nutritional demands or when fed only by D. magna, on which they can complete their development (Sentis, Morisson & Boukal, 2015). Two possible arguments support the latter hypothesis: filter-feeding zooplankters such as D. magna are more likely to be encountered by the S. sanguineum larvae in nature and to have more suitable micronutrient composition as they feed directly on algae (Brett et al., 2017), while Chironomus larvae are mostly buried in sediments and feed on detritus.

Methods

Testing the role of diet conditioning on prey selectivity

In the first experiment, we tested feeding preferences of S. sanguineum larvae offered two prey types in a wide range of ratios. We used representatives of two prey types common at sites inhabited by S. sanguineum: the zooplankter D. magna collected in ponds near České Budějovice and Chironomus larvae (unidentified species) obtained from a local supplier of aquarium fish feed. We tested the hypothesis that preference for the two prey types depends on previous diet of the predators. To this end we collected larvae of S. sanguineum of different instars in a small pond in the south of the Czech Republic (49.1307N, 15.0938E) in May–July 2011, transported them to a climatic chamber at the Institute of Entomology (22 ± 1 °C, 16-h day:8-h night). No permit was required to collect the specimens because the species is not protected by law and the site is in a publicly accessible area not included in any national park or natural reserve. We housed the larvae individually in 80 ml plastic cups (diameter: 57 mm, height: 54 mm) with ca. 70 ml of aged tap water. They were fed by Tubifex worms ad libitum on a daily basis until they reached the penultimate stage.

After the larvae of S. sanguineum reached the penultimate (F-1) instar (n = 124), we relocated them individually into larger plastic containers (15.5 × 10.5 × 10 cm, length × width × height) with cotton fabric glued on the bottom. We assigned each individual randomly in two groups. One half of the individuals were fed ad libitum by D. magna and the other half by Chironomus larvae for 3 days. Afterwards, we left them to starve for 24 h. Observation of feeding behaviour was performed in the same type of plastic containers as those where we kept the larvae prior to the experiment and filled with 200 ml of aged tap water. Shallow water depth (<2 cm) allowed us to record all predation events in the experimental arena despite limited depth of field of our camera (Panasonic HDC-SD90). A single predator was placed into the arena with one of seven different ratios of two prey types (Chironomus:D. magna; 20:0, 16:4, 13:7, 10:10, 7:13, 4:16, 0:20) and feeding was observed and filmed for 30 min. The length of the observation period was chosen based on a pilot experiment which showed that feeding rate declined afterwards due to predator’s satiation. We manually sorted the two prey types prior to the experiment to minimise the variation of prey body size. D. magna used in the experiments had mean body length 2.47 mm (SD = 0.33 mm) and mean dry weight 0.19 mg, while Chironomus larvae had mean body length 11.21 mm (SD = 1.09 mm) and mean dry weight 1.04 mg (SD = 0.37 mg). We carried out 7–10 replicates for each prey ratio. Each predator was used only once.

During the experiment, we recorded each predation event and replaced each prey individual killed by the predator to keep the amount of both prey types constant. We obtained detailed data on each predation event from the recordings using EthoLog 2.2.5 (Ottoni, 2000). We counted the number of approaches towards the prey, attacks, successful attacks, and measured handling time. The predators remained motionless until the initiation of the predation sequence, which prevented us from measuring the encounter rate in a meaningful way. The first stage of the predation sequence we could reliably identify was the approach towards the prey, which we defined as turning of the predator’s body towards the prey. Attack was defined as extending of the dragonfly’s labium, and capture success as the number of captured prey individuals divided by the number of attacks. Handling time was defined as the time from successful attack to mandibular movements’ termination.

To evaluate the selectivity of S. sanguineum larvae towards the two prey types, we calculated Manly’s α selectivity index (Manly, 1974; Chesson, 1983). The index compares the proportion of a prey type in the diet with its proportion in the environment. We used a simple version of the index which assumes that prey abundance in the environment is constant, since we replenished any consumed prey (Manly, 1974). In this case, the formula is:α^i=ri/ni∑j=1mrj/nj, i=1 ,. . . , m

where ri is the number of items of prey i consumed, ni is the abundance of prey type i in the environment, and m is the number of prey types. For the purpose of visualisation, we transformed Manly’s α according to Chesson (1983), so that the value of 0 corresponds to prey consumption identical to prey availability in the environment, positive values correspond to preference for the given prey type, and negative values correspond to avoidance of the given prey type. Since we had only two prey types, we focused on the selectivity towards D. magna in these analyses; preference for D. magna means equally strong avoidance of Chironomus larvae and vice versa.

We tested the effect of diet conditioning experienced by the predator prior to the experiment and the effect of prey availability (the proportion of D. magna) on several measures of foraging behaviour and prey choice. We used generalised linear models (GLM) in R 3.4.4 (R Core Team, 2018), where both predictors (diet conditioning and the proportion of D. magna) and their interaction were included without performing model selection. We fitted separate models for the following dependent variables: the number of D. magna and Chironomus larvae consumed, the number of approaches towards each prey type, the probability of attack, the probability of capture, handling time, and total prey biomass consumed estimated by multiplying the number of prey individuals consumed by the mean dry mass per individual D. magna and Chironomus larva with the assumption that the prey was completely consumed. We chose the error distribution according to the properties of the response variable: we used the quasi-Poisson model for the numbers of events to account for overdisperion, quasi-binomial for ratios, and Gamma with log-link function for the biomass consumed and for handling time. In the analysis of handling time, we used generalised mixed effects models (GLMM) implemented in the lme4 package for R (Bates et al., 2015), because we had multiple observations per individual. Hence, we used predator identity as a random factor in a random intercept model. We tested the significance of the effect of diet conditioning on handling time using a likelihood-ratio test to compare a GLMM model with and without diet conditioning as a predictor.

Testing the effect of diet on growth and mortality

The second experiment aimed to test long-term effects of diet composition on survival, growth rate, developmental time and final body size in S. sanguineum. The experiment was carried out from May to July 2016. We collected 66 larvae of S. sanguineum (instars F-2 and F-1) in the same location as those for the previous experiment. The initial mean body mass was 0.058 g (SD = 0.0112) and body length 10.50 mm (SD = 0.755). We reared the larvae in 200 ml plastic cups (diameter: 65 mm, height: 72 mm) filled by ca. 180 ml of aged tap water in a climatic chamber (20 ± 1 °C, 16-h day:8-h night).

We divided the larvae immediately after transfer from the field into three groups at random. We verified that the initial body mass and length were not significantly different in the three groups (F = 0.67, P = 0.52 for body mass and F = 1.62, P = 0.21 for total body length). We fed them ad libitum as follows: one third of the individuals was fed by D. magna, another third by Chironomus larvae, and the last third by a mixture of both prey types. Water in the cups was changed every 4–5 days and the larvae were checked for moulting and emergence daily. We measured body length and head width and weighed each individual 4 days after each larval ecdysis. Our estimate of growth rate was based on the change of wet body mass between instars F-1 and F-0 of repeatedly measured individuals under the assumption of linear growth. We used the duration of the last instar to characterise development rather than complete developmental time because the wild-caught individuals varied in the developmental stage at the beginning of the experiment. Since we knew from previous experiments that the development of the last instar at the experimental temperature takes ca. 20 days (D. Boukal & M. Peroutka, 2012, unpublished data), we put a wooden stick in each cup 10 days after the last ecdysis to enable larvae to climb out of the water before adult emergence. Subsequently, we recorded the weight, total body length, and head width in the adults measured the 3rd day after emergence to shed excess water and clear their guts, while they were kept in 300 ml plastic boxes in the climatic chamber (20 ± 1 °C, 16-h day:8-h night).

We tested the effect of the diet (only D. magna, only Chironomus larvae, and mixed diet) and sex on measures of growth and survival of the S. sanguineum larvae and traits of the adults using GLM in R 3.4.4 (R Core Team, 2018). We used a quasi-binomial model for mortality, success of the imaginal ecdysis, proportion of viable adults, and a GLM with Gamma error distribution and log link function for duration of the last instar, growth rate, and size and mass of larvae and adults.

Results

Prey selectivity

Although we observed frequency-dependent food selection, that is, the number of each prey type eaten increased with their increasing availability in the environment, there was no evidence for switching. The preference of S. sanguineum larvae for D. magna vs. Chironomus larvae based on Manly’s α selectivity index (Fig. 1) did not depend on the relative availability of the two prey types (F4,82 = 0.35, P = 0.84, prey availability used as a categorical variable), but was significantly affected by diet conditioning (F1,82 = 4.34, P = 0.008). The interaction of the two predictors was not statistically significant (F4,7 = 1.65, P = 0.17). Larvae of S. sanguineum conditioned to D. magna consumed both prey types in agreement with their relative availability (no difference of the selectivity index from random expectation; t42, P = 0.95). However, D. magna was over-represented and Chironomus larvae were under-represented in the diet of S. sanguineum conditioned to Chironomus (selectivity index differed from random expectation; t43, P = 0.0002), see Fig. 1.

Figure 1 Selectivity of larvae of Sympetrum sanguineum towards Daphnia magna.

Values of Manly’s α selectivity index (Manly, 1974) transformed according to Chesson (1983) are plotted to evaluate the selectivity of S. sanguineum larvae towards the two prey types. Values of the selectivity index >0 indicate preference for D. magna, while values <0 indicate avoidance of D. magna.

The difference in prey selectivity between S. sanguineum conditioned to D. magna or Chironomus larvae (Fig. 1) was driven by different consumption of Chironomus larvae. While the number of D. magna consumed by S. sanguineum did not depend on diet conditioning (Fig. 2A, F1,102 = 0.0046, P = 0.95), S. sanguineum conditioned to Chironomus larvae consumed significantly less Chironomus in the experiment across all combinations of prey availability (Fig. 2B, F1,104 = 5.25, P = 0.024). The proportion of Chironomus in their diet was thus reduced and the relative importance of D. magna increased (Fig. 1). Another consequence of the avoidance of Chironomus larvae by S. sanguineum conditioned to them was that their total prey biomass consumption was lower compared to individuals conditioned to D. magna (Fig. 3, F1,121 = 7.39, P = 0.008).

Figure 2 Predation on Daphnia magna and Chironomus larvae by larvae of Sympetrum sanguineum.

Number of D. magna in the diet increased (A) and number of Chironomus larvae decreased (B) with increasing proportion of D. magna in the environment. Number of Chironomus larvae consumed was significantly lower in the larvae of S. sanguineum conditioned to Chironomus larvae prior to the experiment compared to those conditioned to D. magna (B). Number of approaches towards D. magna was independent of diet conditioning (C), while the number of approaches towards Chironomus larvae was lower in predators conditioned to them (D). Point size is proportional to the number of observations with the same x- and y-values. Coloured bands denote the standard error of the predicted values.

Figure 3 Total biomass of prey consumed was lower in larvae of Sympetrum sanguineum conditioned to Chironomus larvae prior to the experiment.

We estimated total biomass of prey consumed by multiplying the number of prey individuals consumed by the mean dry mass per individual D. magna and Chironomus larva with the assumption that the prey was completely consumed. Point size is proportional to the number of observations with the same x- and y-values. Coloured bands denote the standard error of the predicted values.

Analyses of individual steps of the predation sequence showed that the effect of diet conditioning was manifested when the dragonfly larvae approached towards prey. The number of approaches towards D. magna was independent of diet conditioning (Fig. 2C, F1,102 = 0.43, P = 0.51), while the number of approaches towards Chironomus larvae was significantly lower in S. sanguineum conditioned to Chironomus (Fig. 2D, F1,104 = 6.31, P = 0.014). The following steps of the predation sequence were not significantly affected by diet conditioning: probability of attacking D. magna (F1,89 = 2.04, P = 0.16) and Chironomus larvae (F1,77 = 0.03, P = 0.86), capture probability of D. magna (F1,87 = 2.67, P = 0.11) and Chironomus larvae (F1,74 = 2.14, P = 0.15), and handling time of D. magna (χ12 = 0.24, P = 0.62) and Chironomus larvae (χ12 = 1.15, P = 0.29).

The effects of diet on mortality and larval development of Sympetrum sanguineum

Out of the 66 individuals which entered the experiment (20–23 in each treatment), 53 individuals survived until the imaginal ecdysis. Nine individuals died during the imaginal ecdysis, leaving 44 live adults. However, only 38 individuals had no obvious morphological defects (remaining stuck in the larval exuviae, creased wings, deformed legs). Mortality differed significantly between individuals reared on different prey types (Fig. 4A, F2,63 = 3.33, P = 0.042), with Chironomus-only diet leading to the highest mortality (55%). The lowest mortality (17.4%) was found in mixed-diet treatment (Fig. 4A). Data on the probability of successful imaginal ecdysis (defined as the ratio of adults without visible defects and capable of flight to the larvae entering the imaginal ecdysis) mirrored these results, with the lowest success in individuals reared on Chironomus-only diet, and highest success in individuals reared on the mixed diet (Fig. 4B, F2,50 = 5.74, P = 0.006). When larvae of S. sanguineum were reared on Chironomus larvae only, 12 of the initial 20 larvae reached the imaginal ecdysis, during which three died and four other individuals suffered severe defects, leaving only five viable adults capable of flight (i.e. only 25% of the initial number of larvae). Survival to viable adult stage of larvae reared on D. magna-only diet was 2.4-times higher, resulting in 14 viable adults out of 23 larvae (i.e. 61%). Individuals reared on the mixed diet performed best, with 19 viable adults out of the initial 23 larvae (i.e. 83%). These differences in the proportions of viable adults were statistically significant (Fig. 4C, F2,63 = 7.37, P = 0.001).

Figure 4 Mortality and larval development of Sympetrum sanguineum depended on diet.

Mortality (A) refers to the entire period the larvae spent in the experiment until reaching adulthood. The success of the imaginal ecdysis (B) means that the larva entering the imaginal ecdysis emerged as a viable adult capable of flight with no apparent morphological defects. The proportion of larvae emerging as viable adults (C), duration of the last instar (D), and larval growth rate (E) are also shown. Mean and standard error of the fitted values are plotted. Different letters denote statistically significant (P < 0.05) differences between groups based on a post hoc Tukey’s test for multiple comparisons calculated using multcomp package 1.4-8 for R (Hothorn, Bretz & Westfall, 2008).

We also observed differences between diets in the duration of the last instar (Fig. 4D, F2,49 = 32.0, P < 0.0001), which lasted on average 19 days in the mixed-diet treatment, 1 day longer in larvae fed D. magna, and four more days longer in larvae fed Chironomus larvae. Growth rate was also affected by diet (Fig. 4E, F2,47 = 5.52, P = 0.007) and was lowest in S. sanguineum larvae fed only Chironomus.

Although larval diet affected adult size and body mass (Figs. 5D–5F) it did not significantly affect either body mass of F-0 larvae, despite those fed a mixed diet being slightly heavier (Fig. 5A, F2,62 = 2.18, P = 0.12), or the size of F-0 larvae (Figs. 5B and 5C, body length: F2,62 = 0.99, P = 0.38, head width: F2,62 = 0.99, P = 0.38). Adult weight depended on larval diet (Fig. 5D, F2,37 = 3.40, P = 0.044) but did not differ between sexes (F1,35 = 0.58, P = 0.45). Adults from larvae fed only D. magna or only Chironomus had very similar weights, while adults from the mixed diet treatment were heavier. Body length and head width was also affected by larval diet (length: F2,35 = 20.6, P < 0.0001, head width: F2,34 = 12.3, P < 0.0001) but did not differ between sexes (length: F1,33 = 2.15, P = 0.15, head width: F1,32 = 0.82, P = 0.37). Body lengths and head widths of adults from D. magna-only and mixed diet treatments were similar, while adults emerging from larvae fed only Chironomus were ca. 15% shorter and had slightly smaller head widths (Figs. 5E and 5F).

Figure 5 The effect of larval diet on the size of last-instar larvae and adults of Sympetrum sanguineum.

There was no significant difference in body mass (A), body length (B), and head width (C) of last instar larvae, as well as body mass of adults (D), between individuals fed different diets. However, adult body length (E) and head width (F) was smaller in individuals fed only Chironomus larvae. Mean and standard error of the fitted values is plotted. Different letters denote statistically significant (P < 0.05) differences between groups based on a post hoc Tukey’s test for multiple comparisons calculated using multcomp package 1.4-8 for R (Hothorn, Bretz & Westfall, 2008).

Discussion

Our experiments showed that prey identity has important consequences on the effects of previous diet on predator’s feeding preferences, its growth, and survival. We found no evidence for selective feeding in S. sanguineum dragonfly larvae conditioned to D. magna as the larvae consumed D. magna and Chironomus larvae proportionally to their availability in the experiment, while S. sanguineum larvae conditioned to Chironomus apparently avoided Chironomus, which was thus under-represented in their diet. Our previous evidence for S. sanguineum feeding preferences were equivocal: the larvae preferred cladocerans over Chironomus larvae and other alternative prey in one experiment (Klecka & Boukal, 2014), while they fed preferentially on mosquito and Chironomus larvae in a different multiple-choice experiment (Klecka & Boukal, 2012), which demonstrates that prey selectivity is context-dependent. The avoidance of Chironomus larvae by dragonflies conditioned to them is in line with the poor long-term performance of dragonflies reared on Chironomus-only diet. However, S. sanguineum reared on mixed diet had the highest growth rate and survival until adulthood in the long-term experiment. Broader diet thus apparently benefits this predator, as reported in other consumer species (Lefcheck et al., 2013).

What determines predator diet and prey selectivity?

Although prey size is an important predictor of the diet of predators (Woodward & Hildrew, 2002; Brose et al., 2006; Riede et al., 2011; Klecka & Boukal, 2013; Boukal, 2014), it does not explain all variation in prey choice. Under a purely size-dependent optimal foraging perspective, one would predict that S. sanguineum dragonfly larvae should prefer Chironomus larvae that were ca. 5.5-times heavier but had only 2.3-times longer handling time than D. magna. However, this argument relies on the assumption that the two prey types have a similar energetic value per unit mass, which may not be true (Cumminns & Wuycheck, 1971). Multiple factors may affect the energetic value of a prey, such as the proportion of digestible tissue relative to total body mass or prey defence mechanisms (Woodward & Warren, 2007). Prey selectivity may also depend on the interplay between predator’s foraging mode and prey mobility (Allan, Flecker & McClintock, 1987; Sih & Christensen, 2001; Woodward & Hildrew, 2002; Klecka & Boukal, 2012, 2013) and microhabitat use of both prey and predators (Woodward & Hildrew, 2002; Klecka & Boukal, 2012, 2014). However, we performed our experiments in a very simple environment with a limited possibility for these factors to affect the outcomes, although they may be important in natural habitats. Beyond energetics, differences in nutrient composition of the prey may affect the selectivity of the predator depending on its (micro)nutritional demands (Raubenheimer, Simpson & Mayntz, 2009; Wilder & Eubanks, 2010; Brett et al., 2017; Guo et al., 2018).

We performed the experiments on prey selectivity with seven different abundance ratios of the two prey types to also evaluate the effect of relative prey availability on prey selectivity of S. sanguineum larvae. Switching between different prey types based on their relative abundances, specifically preference of the prey type which happens to be more abundant, was observed in some studies (Lawton, Beddington & Bonser, 1974; Sherratt & Harvey, 1993). However, we found no evidence of prey switching in our experiment as the strength of prey preference did not change significantly with relative abundance of the two prey types (Fig. 1).

We expected that predators conditioned to one prey type would either preferentially consume the same prey in the experiment because of an increased detection, capture, or handling efficiency (Bergelson, 1985), or that they would prefer the other prey type to compensate for potential nutritional imbalance caused by a prolonged consumption of a single prey type (Karimi & Folt, 2006; Raubenheimer, Simpson & Mayntz, 2009). Little is known about the ability of dragonfly larvae to learn to capture specific prey or form a search image (Tinbergen, 1960). Bergelson (1985) performed experiments on learning in the larvae of Anax junius (Odonata: Aeschnidae). She found that conditioning to a single prey type led to increased probability of successful capture and decreased handling time, and successful capture reinforced the probability of later attacks on the same prey type. However, there was no indication of a search image formation, that is, no effect of diet conditioning on the probability of orientation towards prey (Bergelson, 1985).

In our experiment, diet conditioning did not affect the attack and capture probabilities or the handling time, and we found no evidence for a positive effect of learning on foraging efficiency. On the contrary, we observed an avoidance of Chironomus larvae by S. sanguineum conditioned to this prey, manifested as lower number of approaches and lower number of Chironomus larvae consumed. This, together with the results of the second long-term rearing experiment, hints at possible compensation for nutritional imbalance and possible lack of essential micronutrients caused by Chironomus-only diet during the 3-day conditioning period (Elser et al., 2000; Fagan et al., 2002; Mayntz et al., 2005; Raubenheimer, Simpson & Mayntz, 2009; Brett et al., 2017) and is comparable to earlier findings of unpalatable prey avoidance in larvae of the coenagrionid damselfly Xanthocnemis zealandica (Rowe, 1994). One limitation of our data is that we cannot estimate the encounter rate, because the predators generally remained motionless until the prey came very close and the first indication that the predator detected the prey was that it moved towards the prey, which we interpret as an approach towards the prey rather than an encounter. We thus rely on comparing the number of approaches towards different prey types rather than on estimating the probability of an approach upon encounter, but consider the conclusions valid.

Does prey selectivity feed back on individual fitness?

Our short-term selectivity experiment and long-term rearing experiment together indicate that long-term fitness consequences rather than short-term energy gains may underlie prey selectivity in larval dragonflies. We observed pronounced long-term effects of diet composition on mortality, growth, and adult size in S. sanguineum. The results are broadly in agreement with the prey selectivity experiment that a Chironomus-only diet may not be suitable for S. sanguineum. Multiple fitness components were affected by the diet. Larvae of S. sanguineum fed only Chironomus had lower survival and growth rate, and lower success of the imaginal ecdysis than those fed only D. magna or a mixed diet. Differences in the initial size of the last-instar larvae were subtle, but the morphology of the adults emerging from larvae reared on Chironomus-only diet was significantly altered: they were shorter and had smaller head width compared to individuals reared on D. magna or mixed diet, although the difference in body masses was small. Interestingly, Hottenbacher & Koch (2006) also reported that larvae of the congeneric S. striolatum reared on Chironomus larvae reached smaller size, measured as head width, compared to larvae reared on Artemia salina, which is not their natural prey. This implies that our results may be valid also for other zooplankton groups and that substantial effects of the diet may be accrued within a single larval instar.

The most likely explanation of the multiple negative effects of Chironomus-only diet on the growth and development of S. sanguineum larvae is based on (micro)nutrient composition of the prey as the predators exposed to the three different diets were fed ad libitum and the small size of the experimental vessels ensured high encounter rate with prey. Although both Chironomidae and Daphniidae are widely used to feed predators in the laboratory, little is known about their exact impact on growth and development of insect predators. Our data do not allow us to provide a definite answer to the question of why the predator individuals fed by Chironomus had higher mortality and slower growth compared to those fed Daphnia or mixed diet. The two prey types differed in multiple ways. Prey size and mobility could affect foraging success of the predator (Klecka & Boukal, 2013), but data from the short-term experiments showed that the predator had high capture success rate and no apparent difficulties in handling either prey type. We are not aware of any indications of toxicity of Chironomus larvae. Unsuitable nutrient composition of Chironomus larvae relative to the predator’s intake target for key (micro)nutrients (Raubenheimer, Simpson & Mayntz, 2009; Brett et al., 2017) remains the most likely explanation. Although we do not have detailed data on nutritional demands of dragonfly larvae and nutrient composition of their prey to properly evaluate the mechanisms underlying their performance on different diets, we can speculate that a possible explanation lies in the availability of algae-derived essential long-chain poly-unsaturated fatty acids accumulated by Daphnia that directly feed on algae in the water column (Brett et al., 2017).

We also found consistent evidence that individuals of S. sanguineum performed better on a mixed rather than D. magna-only diet. This is line with our expectation as dragonfly larvae tend to have broad diets (Klecka & Boukal, 2012). In general, very few predators are strictly specialised, and most benefit from nutritional diversity in their diet (Lefcheck et al., 2013; Brett et al., 2017). Many studies, albeit not on odonates, showed positive effects of prey diversity on growth, survival, and reproduction of other predatory invertebrates such as spiders, beetles, and mites (Oelbermann & Scheu, 2002; Harwood et al., 2009; Muñoz-Cárdenas et al., 2014; Marques et al., 2015). Mixing different prey types in a specific proportion allows predators to reach their intake target for key nutrients (proteins, lipids, etc.) and micronutrients even when individual prey types are not nutritionally balanced relative to the needs of the predator (Raubenheimer, Simpson & Mayntz, 2009; Jensen et al., 2012; Guo et al., 2018). The positive effect of diet diversity may, however, be reversed when the prey mixture contains toxic or nutritionally unsuitable prey (Oelbermann & Scheu, 2002; Lefcheck et al., 2013). However, the superiority of the mixed diet in our study suggests that Chironomus larvae are not directly toxic for S. sanguineum larvae or that any costs of potential toxicity are outweighed by the benefits of mixed diet.

Conclusions

In conclusion, we found that larval diet can significantly affect foraging behaviour, survival and growth of dragonfly larvae and body size of the emerged adults. The combined evidence from our two experiments shows that Chironomus larvae are lower-quality prey for S. sanguineum than D. magna, but also that the predator survives and grows best on a mixed diet. Surprisingly, the effects of diet conditioning on dragonfly foraging behaviour were limited to the avoidance of the inferior prey after previous exposure to it, which suggests some but limited role of learning in their foraging. Our study shows the merit of combining short-term studies on prey selectivity with long-term rearing experiments. Future research should also focus on obtaining detailed insights into (micro)nutritional demands of predators and (micro)nutrient composition of their prey to better understand mechanisms driving prey selectivity in predatory invertebrates.

Supplemental Information

Supplemental Information 1 Data on selective predation from the short term experiment.

Each observation contains information about the number of Daphnia magna and Chironomous larvae offered as prey and the number of approaches towards each prey type, the number of attacks, and the number of succesfully captured prey.

Click here for additional data file.

Supplemental Information 2 Handling times of the larvae of Sympetrum sanguineum processing their prey during the short-term experiment on prey selectivity.

Information about the number of Daphnia magna and Chironomus larvae available and handling time for individual predation events are provided for each dragonfly larva identified by a unique ID.

Click here for additional data file.

Supplemental Information 3 Data from the long-term experiment on the effect of diet on growth and development of Sympetrum sanguineum.

Each line contains information about the survival, body weight, body length, and head width of individual larvae of S. sanguineum, and the same data on the adults in individuals which successfully completed their development.

Click here for additional data file.

We would like to thank two anonymous reviewers for their suggestions which helped improve our manuscript.

Additional Information and Declarations

Competing Interests

Author Contributions

Data Availability

The authors declare that they have no competing interests.

Pavla Dudová conceived and designed the experiments, performed the experiments, analysed the data, contributed reagents/materials/analysis tools, prepared figures and/or tables, authored or reviewed drafts of the paper, approved the final draft.

David S. Boukal conceived and designed the experiments, contributed reagents/materials/analysis tools, authored or reviewed drafts of the paper, approved the final draft.

Jan Klecka conceived and designed the experiments, performed the experiments, analysed the data, contributed reagents/materials/analysis tools, prepared figures and/or tables, authored or reviewed drafts of the paper, approved the final draft.

The following information was supplied regarding data availability:

Raw data from the short-term experiment on prey selectivity are available as Files S1 and S2. Raw data from the experiment on long-term effect of diet composition on growth and survival of the predator are available as File S3.

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
