# Peer review of "Prey selectivity and the effect of diet on growth and development of a dragonfly, Sympetrum sanguineum"

_PeerJ, doi:10.7717/peerj.7881_

## Round 0.1 · original submission · Major Revisions

Dear Dr. Dudová and colleagues:

Thanks for submitting your manuscript to PeerJ. I have now received two independent reviews of your work, and as you will see, the reviewers raised some concerns about the research. Despite this, these reviewers are optimistic about your work and the potential impact it will lend to research on dragonfly biology. Thus, I encourage you to revise your manuscript, accordingly, taking into account all of the concerns raised by both reviewers.

Please ensure that all relevant literature is cited in your revision. Please also make sure the figures are clear and stand-alone. There are many suggestions made by both reviewers regarding clarity, and how the manuscript can be more focused in scope and delivery. Please consider all of this valuable criticism.

Therefore, I am recommending that you revise your manuscript, accordingly, taking into account all of the issues raised by both reviewers. I do believe that your manuscript will be ready for publication once these issues are addressed.

Good luck with your revision,

-joe

Reviewer 1 ·

Basic reporting

No comment.

Experimental design

No comment

Validity of the findings

No comment.

Additional comments

Comments to the Authors:
The authors investigate the effect of prey selectivity and diet on growth and survival in a dragonfly species. Two prey types were presented to the dragonfly larvae, Daphnia magna and Chironomus sp., in two separate trials. Dragonfly larvae consumed less chironomids across different combinations of prey availability. Moreover, metrics for approaches, attacking, capture, and handling were determined (but did not differ). The authors found that dragonflies had lowest mortality and greater mass on the mixed diet.

This study explores an interesting and relatively understudied line of questioning. Many comprehensive aspects of behavioral responses to diet have been explored in this study. My major concerns pertain to invoking perspectives from optimal foraging and ecological stoichiometry. The Introduction section presently offers substantial insight into dragonfly behavior and fairly great amount of effort is placed on outlining open areas for research. However, the section could also benefit from elucidating those nutritional frameworks. At least for a behavioral foraging study, although diets contain energy and elements it is worth noting that energy is a property of macronutrients and that most consumers’ sensory systems are tuned into macronutrients rather than elements (Raubenheimer et al. 2009 Functional Ecology). Further, elements and energy alone do not offer insight into digestibility (Wilder and Eubanks 2010 Ecology).

The nutritive composition of prey consumption in this paper is cited generally as “quality” (e.g. Lines 57-58, 317-320, 342-343). The authors speculate that their mortality and growth findings are driven by phosphorus (Lines 323-326), but this is rather weakly substantiated. The molar ratio of phosphorus was inferred to be twice as high in cladocerans than chironomids, but this could be confounded by the greater mass of chironomids (perhaps greater total magnitude phosphorus), if there were to be selective feeding on different prey parts by dragonflies, and perhaps other factors. The energetics of the consumed prey were incomplete, as only the initial prey mass was presented. To robustly refute any explanation of energetics (optimal foraging), elements (stoichiometry), or macronutrients (geometric framework) in these trials, a table of prey nutrients would be beneficial.

The scope of this study is limited to a single consumer species collected from a single small pond within a single season. In addition to potentially being of different nutritional composition, the prey also differed in one being collected from the field (Daphnia) and the other being purchased from a local supplier (Chironomus). Competing hypotheses of prey diet effects are so far limited to a relatively high mass low phosphorus prey (Chironomus) and low mass high phosphorus prey (Daphnia). It is possible that other prey-specific factors contributed to predator growth differences (e.g. relative digestible energy, protein, chitin, etc.), but the information on these are not yet available in this paper.

Specific Comments:
1. Line 57-58: The phrases “right quality” and “resource” in this paragraph should be more nutritionally explicit. That is, predators forage for prey containing different elements (particularly C, N, P), which form the macronutrients structures selected by consumer sensory systems and those macronutrients can also in turn be combusted for energy. The explicit nutrient is highly pertinent for disentangling behavioral and physiological responses to diet.
2. Lines 93-102: From this draft, the reader cannot yet determine if the prey is nutritionally balanced (i.e. prey composition) or what the nutritional demands of dragonflies are, which potentially weakens the proposed hypotheses.
3. Lines 103- : The Methods would be strengthened/condensed with a table for the prey composition (i.e. masses, element molar ratios, macronutrients, and energetics). At present, the only possible dietary comparison between the prey types from the Methods section is total dry mass.
4. Line 119: Fully randomized diet grouping was likely more effective than a lack of pre-determined scheme, but a sized-matched randomization scheme (i.e. aligning predator masses or lengths) could perhaps have more fully interspersed size-dependent variation across treatments.
5. Lines 168-170: Similar to the previous comment, it is not yet apparent that the variation in sizes and/or developmental stages of dragonfly larvae were interspersed between the treatment groups.
6. Lines 259-270: Pertaining to optimal foraging and energetics, besides mass it is also likely that the Daphnia magna contained a greater relative amount of energy than Chironomus (e.g. Cummins and Wuycheck 1971 Int. Ver. Theor. Angew. Limno. Verh.)
7. Lines 323-326: Given that the two prey types were effectively equivalent in most predation metrics following conditioning, the influence of nutrients seems to be particularly important in seeking explanations for long-term growth and mortality differences. The authors seem to be referencing the growth rate hypothesis (Elser et al. 1996 BioScience) when outlining the C:N:P of prey and this could explain growth differences between the Daphnia and Chironomus populations. However, this framework offers limited insight into predator growth without some understanding of the predator’s body composition and/or recycling.
8. Lines 326-328: This statement is broadly speculative. One aspect of predator fitness was investigated in this study (i.e. growth), but predictions of predator versus herbivore/prey nutrient composition have not yet been explicitly outlined (e.g. higher nitrogen composition). A series of counter-arguments against the frequently proposed predator composition/stoichiometry driving dietary selection paradigm were also provided in Wilder and Eubanks 2010 Ecology.

Reviewer 2 ·

Basic reporting

The authors investigate the links between short-term prey selectivity and long-term effects on fitness-related traits using S. sanguineum larvae.

The authors use clear, unambiguous, and professional language throughout while providing sufficient lit. references relative to the field and the study.
The authors results are relevant to the hypotheses; however, I have some confusion in how they interpret some of these results with regard to the prey selection experiment which should be addressed. I explain my confusion in more detail in later sections.

Experimental design

The research question is well defined. Experimental design is sound. It is stated how research fills an identified knowledge gap, especially with regards to blending short term prey selection studies and tracking the long-term effects of this diet selection. Overall, the methods described with sufficient detail, however, I have some additional questions that should be addressed:

Did you standardize the size or age of the prey? If so, explain. If not, state you did not.

Is it a selectivity index or electivity index? Both terms are used in text and on figures.

Line 172 calculated growth rate based on change in mass. Was this a dry weight on the different individuals or wet weights measured on the same individuals? You should state which is the case.

Fig. 2 The proportion of Chironomus larvae and Daphnia (the the x-axes) and different between Fig 2 A & B and Fig 2 C & D. Is the minimum proportion 0.0 or 0.2? The maximum proportion 0.8 or 1.0?

Fig. 3 how did you get dry mass of the prey consumed?

Could not find a reference to Fig. 3 in the text. If you don’t reference a figure in the text, is it important enough to be included in the main document? Perhaps in the supplementary material, or be sure to reference the figure.

Fig 4 & 5 post hoc analysis to show significant differences between treatment (labeled with letters above the data point on each figure) would be helpful in interpreting the results.

Validity of the findings

Conclusions are well stated, linked to original research questions and limited to supporting results. However, I am somewhat confused with how the authors are interpreting the prey selectivity results:

Line 195 The authors state S. sanguineum consumed more Daphnia and less Chironomus than expected based on prey availability referencing Figure 1. However, figure 2 A shows no difference in the number of Daphnia consumed between the two diet conditioning treatments. How can both of these be true? Again, the only significant results I can glean from figure 2 is that Chironomus conditioned S. sanguineum consume less Chironomus than S. sanguineum conditioned to Daphnia. Maybe I'm missing something, if so, the interpretation needs to be more clear.

Line 249 I’m not sure you can say that the S. sanguineum conditioned to Chrionomus consistently preferred D. magna, as they ate the same amount of Daphnia as the S. sanguineum conditioned to Daphnia. S. sanguineum conditioned to Chrionomus simply just ate less Chironomus. Is this not what you show in Figure 2? -- # of Daphnia consumed is not different (Fig 2A); the Chironomus conditioned treatment eat less Chironomus (Fig 2B).

Additional comments

Abstract line 26. “no effect of experience” you mean diet here, correct? Maybe rephrase

Line 29 “performed significantly worse” Line 30 “performed the best” I would rephrase to had significantly lower growth and survival or the highest growth survival rather than “worst” or “best.” This may be a trivial point but the current language does not sound particularly scientific.

Line 42 – 46 is confusing, perhaps reword to something like, “Earlier research centred around the concept of optimal foraging theory (Emlen, 1966; MacArthur and Pianka, 1966; Stephens and Krebs, 1986) focused on the importance of energy gains on prey selectivity, positing that consumers should maximise their energy intake by selectively consuming the most profitable resource, i.e., a resource which provides the highest energy intake per unit of time.”

Line 150 Authors use “experience” and “diet conditioning” interchangeably throughout the paper, are these terms different? If not, choose one. I prefer diet conditioning as it is more descriptive.

---

## Round 0.2 · accepted · Accept

Dear Dr. Dudová and colleagues:

Thanks for re-submitting your revised manuscript to PeerJ, and for addressing the concerns raised by the reviewers. I now believe that your manuscript is suitable for publication. Congratulations! I look forward to seeing this work in print, and I anticipate it being an important resource for research communities studying dragonfly biology.

Thanks again for choosing PeerJ to publish such important work.

-joe

Reviewer 1 ·

Basic reporting

No comment.

Experimental design

No comment.

Validity of the findings

No comment.

Additional comments

The authors appear to have contributed substantial revisions since the initial round of review. That is, the authors seemed to provide much greater insight into frameworks pertinent to foraging ecology (i.e. optimal foraging, ecological stoichiometry, and the geometric framework). Moreover, the authors appear to sufficiently address concerns pertaining to experimental design, such as the size matching schema.

Inferences on the mechanisms driving the differences in predator fitness (e.g. mortality and growth rate) are still limited to differences in prey species identity and mass. Elemental, micro/macro-nutritional, and/or energetic direct measures or estimates from the literature are not yet available in the manuscript. Revisions in the Discussion and Conclusion sections reflect this concern and highlight the significance of future research delving into these potential mechanistic drivers of predator diet mixing. Again, the authors have provided significant revisions to this interesting, potentially impactful paper and I look forward reading upcoming research further disentangling prey selection and dietary effects.

Reviewer 2 ·

Basic reporting

no comment

Experimental design

no comment

Validity of the findings

no comment

Additional comments

The authors have revised and addressed the concerns of both authors. Therefore I recommend accepting for publication.